# aYAP modRNA reduces cardiac inflammation and hypertrophy in a murine ischemia-reperfusion model

Jinmiao Chen[1,2,*], Qing Ma[1,*], Justin S King[1], Yan Sun[3], Bing Xu[3], Xiaoyu Zhang[1], Sylvia Zohrabian[1], Haipeng Guo[1,4], Wenqing Cai[5], Gavin Li[1], Ivone Bruno[6], John P Cooke[6], Chunsheng Wang[2], Maria Kontaridis[3], Da-Zhi Wang[1], Hongbo Luo[1], William T Pu[1,7], Zhiqiang Lin[1,3]

**Myocardial recovery from ischemia-reperfusion (IR) is shaped by the interaction of many signaling pathways and tissue repair processes, including the innate immune response. We and others previously showed that sustained expression of the transcriptional co-activator yes-associated protein (YAP) improves survival and myocardial outcome after myocardial infarction. Here, we asked whether transient YAP expression would improve myocardial outcome after IR injury. After IR, we transiently activated YAP in the myocardium with modified mRNA encoding a constitutively active form of YAP (aYAP modRNA). Histological studies 2 d after IR showed that aYAP modRNA reduced cardiomyocyte (CM) necrosis and neutrophil infiltration. 4 wk after IR, aYAP modRNA–treated mice had better heart function as well as reduced scar size and hypertrophic remodeling. In cultured neonatal and adult CMs, YAP attenuated $H_2O_2$- or LPS-induced CM necrosis. TLR signaling pathway components important for innate immune responses were suppressed by YAP/TEAD1. In summary, our findings demonstrate that aYAP modRNA treatment reduces CM necrosis, cardiac inflammation, and hypertrophic remodeling after IR stress.**

## Introduction

Coronary artery disease is one of the leading causes of morbidity and mortality in industrialized countries. Although early percutaneous coronary intervention strategies have improved survival in acute myocardial infarction (MI) patients (Sugiyama et al, 2015), survivors remain at risk of developing heart failure. Restoration of blood flow to the ischemic heart unavoidably causes additional injury to the myocardium (Lønborg, 2015). The causes of reperfusion injury are multifactorial, including the influx of reactive oxygen species, calcium overload, inflammation, and capillary dysfunction (Prasad et al, 2009). After percutaneous coronary intervention, increased inflammatory markers, such as the circulating neutrophil count and serum C-reactive protein level, are associated with poor prognosis (Chia et al, 2009; Reindl et al, 2017). Animal studies suggest that dysregulation of inflammatory responses after myocardial injury results in worse remodeling (Nahrendorf et al, 2007; van Amerongen et al, 2007).

The immune response can be divided into two arms, innate and adaptive, that are both involved in the cardiac injury response. After ischemia-reperfusion (IR) injury, innate immune cells are first recruited into the injured heart, followed by the activation of adaptive immune cells (Epelman et al, 2015). These immune cells coordinate the removal of dead and damaged cells, clearing of extracellular matrix debris, revascularization, and scar formation (Zlatanova et al, 2016). Signals in the injured myocardium that attract innate immune leukocytes include death/damage-associated molecular patterns (DAMPs), cytokines, and chemokines (Zlatanova et al, 2016). TLRs are one type of pattern recognition receptors that sense DAMPs (Timmers et al, 2012). With the help of accessory proteins, such as CD14 (Lee et al, 2012), TLRs bind DAMPs and activate downstream signaling cascades to promote the expression of cytokines and chemokines (Timmers et al, 2012) and to trigger programmed cell necrosis (He et al, 2011). In the human heart, the expression of TLRs 1-10 has been detected, with TLR2 and TLR4 being the most abundant (Nishimura & Naito, 2005). In mice, whole body inactivation of TLR4 reduced myocardial inflammation and infarct size after IR stress (Chong et al, 2004).

The Hippo-yes-associated protein (YAP) pathway was originally discovered because it is essential for organ size control (Pan, 2010). YAP, a transcriptional co-activator that promotes cell proliferation and survival, is inhibited by a series of kinases, such as Mst1/2-Sav kinase complex and Lats1/2 kinase. This pathway is essential for heart

---

[1]Boston Children's Hospital, Boston, MA, USA   [2]Department of Cardiovascular Surgery and Shanghai Institute of Cardiovascular Disease, Zhongshan Hospital, Fudan University, Shanghai, China   [3]Masonic Medical Research Institute, Utica, NY, USA   [4]Department of Critical Care Medicine, Key Laboratory of Cardiovascular Remodeling and Function Research, Chinese Ministry of Education and Chinese Ministry of Health, Qilu Hospital of Shandong University, Jinan, China   [5]Boston Children's Hospital and Dana Farber Cancer Institute, Boston, MA, USA   [6]Houston Methodist Research Institute, Houston, TX, USA   [7]Harvard Stem Cell Institute, Cambridge, MA, USA

Correspondence: zlin@mmri.edu; wpu@pulab.org
*Jinmiao Chen and Qing Ma contributed equally to this work

development and cardiac regeneration (Xiao et al, 2016; Liu & Martin, 2019). In murine MI models, activation of YAP improves cardiac regeneration and reduces myocardial scar size (Xin et al, 2013; Lin et al, 2014). SAV is a scaffold protein that forms a complex with Mst1/2 kinase to activate Lats1/2. Inactivation of SAV promotes cardiac regeneration and reverses cardiac systolic function post-MI (Heallen et al, 2013; Leach et al, 2017).

Recent studies have begun to identify a role of Hippo-YAP in the regulation of immune responses (Hong et al, 2018). In Drosophila, bacterial infection activated Hippo kinases via TLR signaling, and genetic perturbation of the Hippo-YAP pathway increased sensitivity to bacterial infection (Liu et al, 2016). In mouse, depletion of YAP in macrophages enhanced innate immune responses, decreased virus infection, and improved mouse survival (Wang et al, 2017). In the heart, depletion of YAP/TAZ (transcriptional co-activator with PDZ-binding motif) in the epicardium exacerbated cardiac inflammatory responses and worsened outcome after MI, probably because of reduced recruitment of Tregs, a subset of adaptive immune cells (Ramjee et al, 2017).

Our group and others previously showed that sustained YAP activation after MI improves myocardial outcome and increases mouse survival (Xin et al, 2013; Lin et al, 2014). Heterozygous deletion of Yap increased CM death and scar size after MI stress (Del Re et al, 2013), further supporting the importance of YAP in myocardial responses to injury. The risk of oncogenicity makes sustained YAP expression untenable as a translational therapy. Given the potential roles of YAP in regulating cell survival and innate immune responses, we hypothesized that transient activation of YAP in the myocardium after IR injury would be cardioprotective. Here, we tested this hypothesis by using modified mRNA (modRNA) (Zangi et al, 2013) to transiently express constitutively active human YAP (aYAP modRNA) in mouse hearts after IR injury. Our data showed that aYAP modRNA improved heart function, reduced scar size, and prevented hypertrophic cardiac remodeling. Mechanically, aYAP modRNA decreased CM necrosis and attenuated inflammatory innate immune responses.

## Results

### Intramyocardial delivery of aYAP modRNA

Compared with regular mRNA, modRNA is more stable and less immunogenic (Sahin et al, 2014). In this study, the YAP modRNA encodes FLAG-tagged human YAP containing a serine 127 to alanine mutation, which activates YAP (aYAP) by making it resistant to Hippo kinase phosphorylation (Lin et al, 2014). The aYAP modRNA was HPLC-purified after in vitro transcription (Fig S1A). We first validated that aYAP modRNA successfully expressed YAP in the cultured neonatal rat ventricular myocytes (NRVMs) (Fig 1A). Then, we checked aYAP modRNA expression in the myocardium. In all of the following studies, we directly injected 50 μg aYAP modRNA into the border zone of the ischemic region shortly after left anterior descending coronary artery (LAD) ligation. The same volume of vehicle (saline) was injected as a control (Veh group). 2 d after IR, hearts were collected for analysis. aYAP was successfully expressed in the myocardium, as demonstrated by qRT-PCR, Western blot, and immunohistochemistry (Fig 1B–D). CMs were successfully transduced by aYAP modRNA (Fig 1E). In transduced CMs, aYAP was detected in both cytoplasm and nucleus (Fig 1F). Other cell types, such as endothelial cells, were also transduced (Fig S1B). These

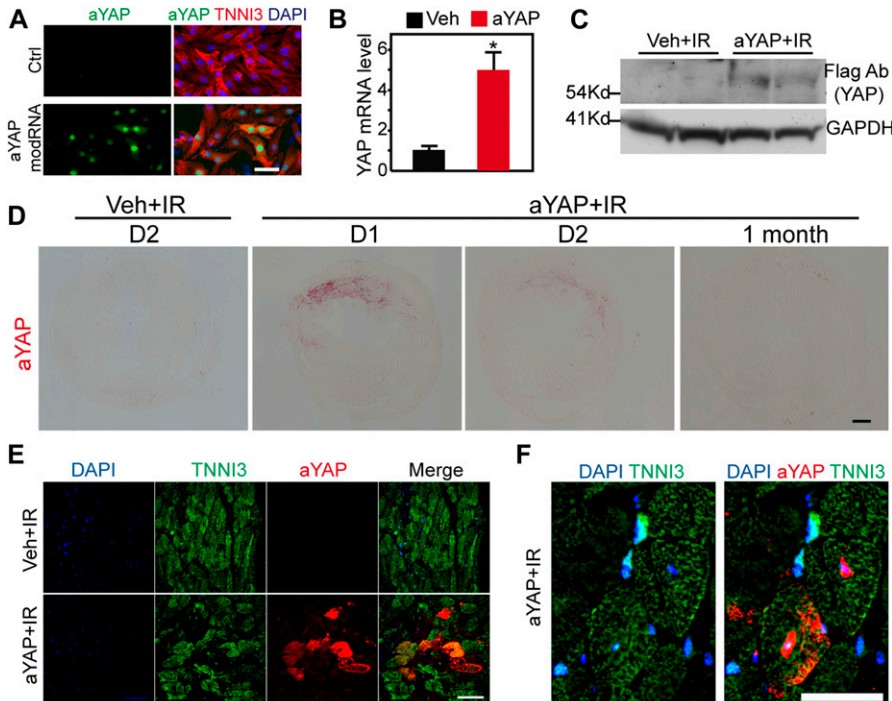

**Figure 1. aYAP modRNA expression in the myocardium.**
**(A)** aYAP modRNA–transduced cardiomyocytes. 16 h after aYAP modRNA transfection, NRVMs were fixed for immunofluorescence staining. Bar = 50 μm. **(B)** YAP mRNA level measured by qRT-PCR. *P < 0.05, n = 3. Hearts were collected 2 d after IR and modRNA injection. **(C)** Immunoblot to show the expression of YAP protein. aYAP modRNA was fused to a 3×Flag tag. Flag antibody immunoblot showed the expression of aYAP in the aYAP modRNA–treated group but not in the vehicle-treated group. **(D)** Immunohistochemistry staining of Flag-YAP one day (D1), two days (D2) and 1 mo after IR. Bar = 500 μm. **(E, F)** Immunofluorescence staining of Flag-YAP. Cardiac troponin I (TNNI3) was used to label cardiomyocytes. **(E)** Low magnification of vehicle or aYAP-transduced myocardium. Bar = 50 μm. **(F)** high magnification of aYAP-transduced cardiomyocytes. Bar = 25 μm.

results demonstrate that aYAP modRNA expresses YAP in the myocardium.

## aYAP modRNA reduces cell death after IR

Sustained, genetic activation of YAP increases CM proliferation and reduces cell death after myocardial injury (Xin et al, 2013; Lin et al, 2014). We tested whether transient YAP activation mediated by aYAP modRNA had similar effects. The experimental protocol is shown in Fig 2A. The LAD was occluded, and aYAP modRNA (aYAP) or saline vehicle (Veh) was injected at three sites of the ischemic border region. Red fluorescence microbeads were injected into the left ventricular cavity to label the myocardial regions perfused by the coronary arteries (Hale et al, 1986). After 50 min, the LAD ligature was released. 1 d after IR, anti-myosin (MF20) antibody was also injected into the mice to label necrotic CMs with membranes that were permeable to the injected antibody (Lin et al, 2016). The mice were euthanized on the second day after IR.

We analyzed histological sections to assess the effect of aYAP modRNA on the myocardial injury response. To make sure that mice in both treatment groups received comparable injuries, we measured the area at risk (AAR), that is, the fractional myocardial area that lacked red fluorescent microbeads because of LAD occlusion. The AAR was indistinguishable between Veh+IR and aYAP+IR groups (Fig S2A and B). Next, we measured CM proliferation and death. Using cardiac troponin I (TNNI3) as a CM marker, we separated the myocardium into three zones: infarct myocardium, lacking TNNI3 signal; border zone myocardium, containing TNNI3+ myocardium as well as nonviable myocardium; and remote myocardium, containing viable TNNI3+ myocardium that was distant from the infarct (Fig S2C).

We tested whether aYAP modRNA reduced cell death. Terminal deoxynucleotidyl transferase (TdT) dUTP Nick-End Labeling (TUNEL) staining was used to label apoptotic cells. CMs with TUNEL signals in aYAP+IR hearts were less than that of the Veh+IR hearts (Fig S2F and G), but this difference did not reach statistical significance. To study CM necrosis, we identified CMs with membranes sufficiently permeable to allow entry of anti-myosin MF20 antibody. Compared with Veh+IR, aYAP+IR hearts had significantly less MF20-positive myocardial area (20.2% ± 5.96% versus 31.3% ± 6.08%, $P = 0.031$, Fig 2B and C), suggesting that aYAP reduces CM necrosis. We did triphenyltetrazolium chloride staining to further check myocardial viability 2 d after IR. Myocardial infarct size was significantly smaller in mice treated with aYAP modRNA compared with control treatment (Fig 2D and E), confirming that aYAP modRNA reduced cardiomyocyte necrosis. Consistent with these results, serum cardiac troponin T (cTnT), a diagnostic marker for CM death in patients with MI (Daubert & Jeremias, 2010), was significantly lower in aYAP+IR than Veh+IR at 2 d, but not 1 d, after IR (1.49 ± 0.1 versus 2.75 ± 0.79 ng/ml, $P = 0.0485$, Fig 2F). Together, these data indicate that aYAP modRNA increases CM survival after IR injury.

## aYAP modRNA reduces myocardium neutrophil infiltration

YAP suppressed innate immune responses under pathogen stress in both Drosophila (Liu et al, 2016) and mouse (Wang et al, 2017). We,

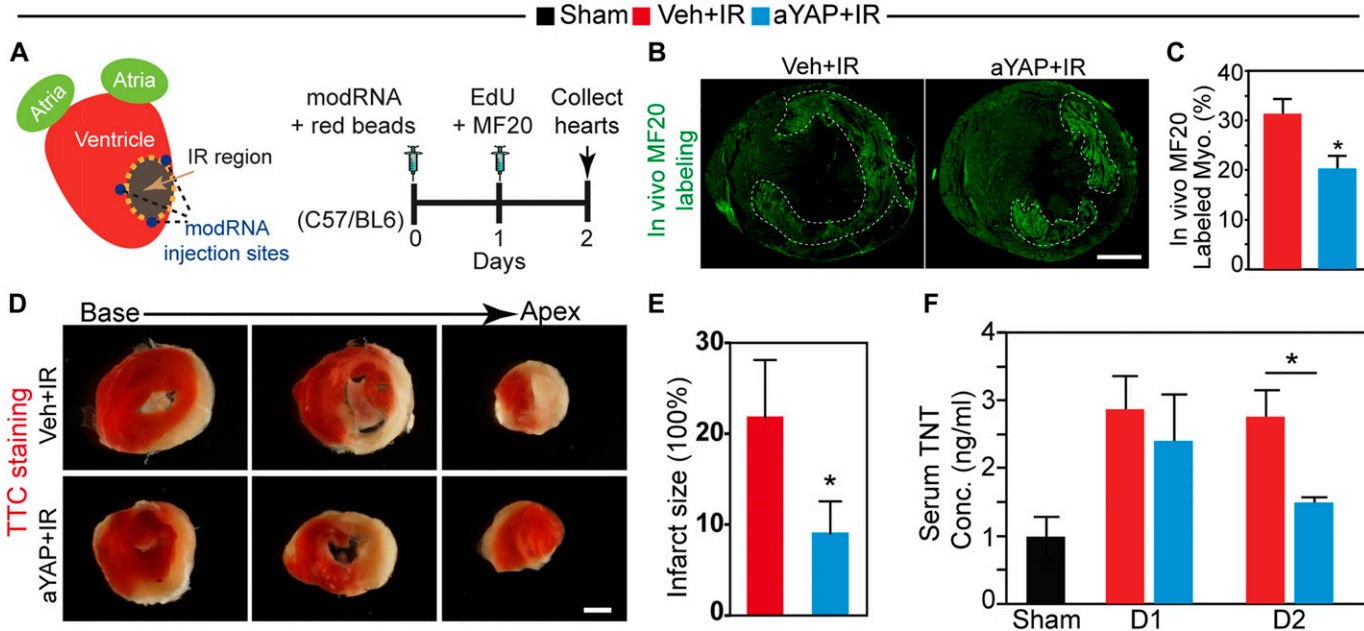

**Figure 2. aYAP modRNA increases CM survival.**
**(A)** Experimental design. IR and modRNA injection was performed as described in the Materials and Methods section. Red fluorescence beads were injected into the left ventricle cavity after LAD ligation to label blood perfused area. 1 d after IR, EdU and MF20 antibodies were intraperitoneally injected to label CMs proliferation and necrosis, respectively. At day 2 after IR, hearts were collected for histology analysis. **(B)** MF20 staining of heart cross sections. Bar = 500 μm. **(C)** Quantification of myocardium with MF20 signals. t test: *P < 0.05, n = 4. **(D)** Triphenyltetrazolium chloride staining of heart cross sections 2 d after IR. Bar = 200 μm. **(E)** Quantification of myocardial scar size. t test: *P < 0.05, n = 4. **(F)** Peripheral blood serum troponin T concentration at different days after IR and aYAP modRNA treatment. Sham n = 3; Veh+IR, n = 4; aYAP+IR. t test: *P < 0.05.

therefore, tested whether aYAP modulates the innate immune response after IR stress. We analyzed cardiac inflammation at one (D1) or two (D2) days after IR and aYAP modRNA treatment. Mice that underwent surgery but not LAD ligation and modRNA treatment were used as sham controls. Basal and apical portions of the heart were used for histological and FACS, respectively. Histological analysis of D1 myocardium showed that both Veh+IR and aYAP+IR hearts had significantly higher non-CM density than sham, and aYAP+IR tended to have lower non-CM density than Veh+IR (Fig S3A). We then measured myocardial infiltration of neutrophils and monocytes/macrophages by staining for Ly6G and Mac-3, respectively (Lavine et al, 2014). Both Ly6G[+] and Mac-3[+] cells were rare in sham, and their densities were markedly higher in the IR groups (Fig S3B and C). Compared with Veh+IR, aYAP+IR hearts had similar Ly6G[+] cell density and higher Mac-3[+] cell density (Fig S3B and C). On D2, aYAP+IR had much lower non-CM density than the Veh+IR (Fig 3A and B). On D2, both Veh+IR and aYAP+IR, but not sham, had obvious Ly6G signals in the myocardium (Fig 3C). In comparison with Veh+IR, aYAP+IR had much lower Ly6G[+] myocardial area fraction (12.53% ± 5.67% versus 29.81% ± 11.02%, P = 0.0435; Fig 3C and D) and similar Ly6G[+] cell density (Fig 3E and F), suggesting that aYAP primarily reduces the size of the affected region. Unlike D1, the Mac-3[+] density in the aYAP+IR group was lower than in the Veh+IR, but did not reach statistical significance because of intragroup variation (Fig S3D). These data suggest that aYAP modRNA reduces the infiltration of neutrophils 2 d after IR.

To confirm the findings from the histological studies, we performed flow cytometry on non-CMs dissociated from D2 heart apex. Single non-CM cells were first separated by their expression of CD45, a marker of the hematopoietic lineage (Fig S3E). CD45[+] myeloid lineage cells were further separated into neutrophils (CD11b[+];Ly6G[+]) and macrophages (F4/80[+];CD11b[+]). Compared with sham, both Veh+IR and aYAP+IR had more CD45[+] cells in the myocardium. Compared with Veh+IR, aYAP+IR had strikingly less CD45[+] cells than the Veh+IR (40.2 ± 11.8 versus 14.5 ± 2.6 cells/μg, P = 0.0073; Figs 3G and S3F) and also much less neutrophils (14.9 ± 2.6 versus 6.4 ± 1.5 cells/μg, P = 0.0023; Figs 3H and S3G). There were also less macrophages in the aYAP+IR hearts than in the Veh+IR hearts, although this difference was not statistically significant (Figs 3I and S3H).

In addition to the heart, we also measured circulating white blood cell counts to look for systemic changes. On D1, the white blood cell, red blood cell, neutrophil, and lymphocyte blood counts were not significantly different between sham, Veh+IR, and aYAP+IR. On D2, white blood cells, neutrophils, and lymphocytes were not significantly different between sham and Veh+IR. However, compared with Veh+IR, aYAP+IR mice had significantly lower white blood cells, neutrophils, and lymphocyte counts (Fig 3J–L). As a quality control, the red blood cell concentration was similar between these groups at both D1 and D2 (Fig 3M).

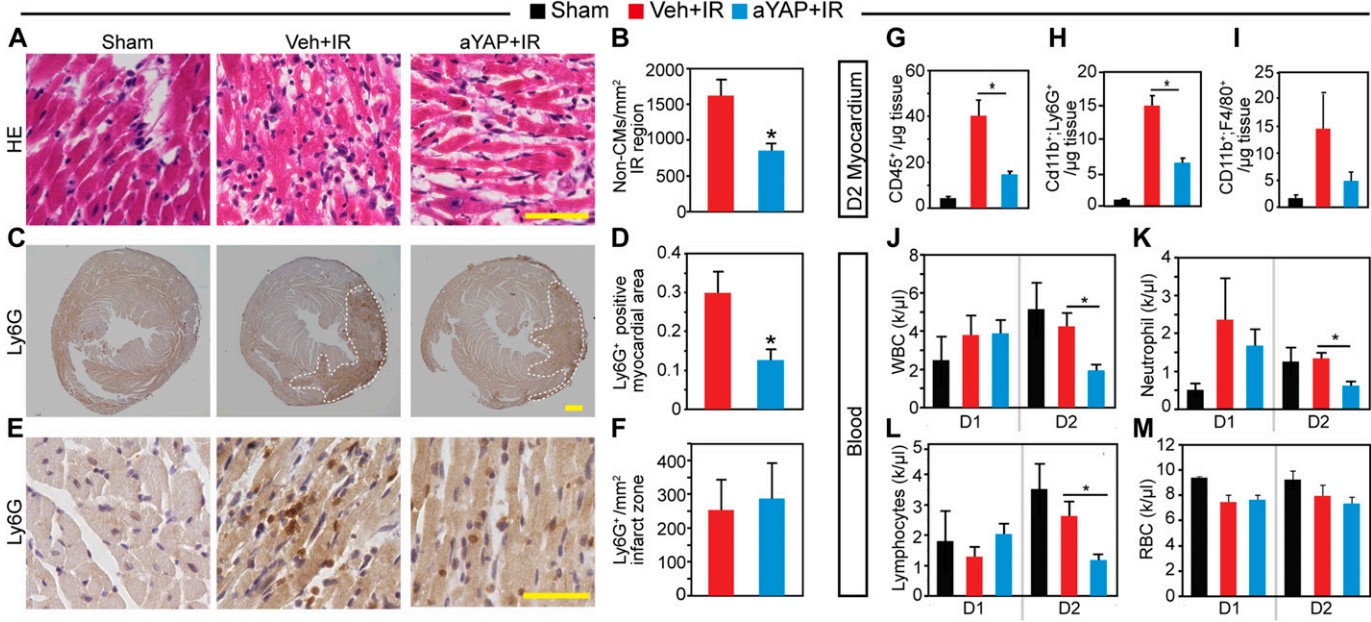

**Figure 3. aYAP modRNA reduces cardiac inflammation after IR.**
**(A)** Representative images of hematoxylin and eosinstained heart sections. In Veh+IR and aYAP+IR, typical images from the ischemia regions were shown. Bar = 50 μm. **(B)** Quantification of non-CMs in the infarct region. t test: *P < 0.05, n = 4. **(C)** Heart sections stained with Ly6G antibody, low magnification. White dotted lines indicate Ly6G[+] myocardium. Bar = 500 μm. **(D)** Quantification of Ly6G[+] myocardium. Ly6G[+] myocardium area was normalized against the whole myocardium area. t test: *P < 0.05, n = 4. **(E)** High magnification of heart sections stained with Ly6G antibody. Images were taken from infarct zone. Bar = 50 μm. **(F)** Ly6G[+] cell density in the infarct zone. **(A, B, C, D, E, F)** Heart sections from day 2 (D2) after IR were used. **(G, H, I)** Quantification of different myeloid lineage leukocytes. Myocardiums from D2 were dissociated and non-CMs were enriched. For flow cytometry analysis, non-CMs were stained with indicated antibodies. Myeloid lineage leukocytes were labeled with CD45 antibody. CD45[+] cells were further separated into neutrophils (CD11b[+]; Ly6G[+]) and macrophages/monocytes (CD11b[+]; F4/80[+]). Cell number was normalized to the myocardium weight. One-way ANOVA post hoc tests. N = 4. *P < 0.05. **(J, K, L, M)** Peripheral blood cell composition at day 1 (D1) and day 2 (D2) after IR. Blood samples from affected mice were collected and analyzed on a HEMAVET 950FS auto blood analyzer. The counts of white blood cells, red blood cells, neutrophils, and lymphocytes were automatically analyzed. k/μl, 1,000 cells per microliter blood. Sham, n = 3. Veh+IR, n = 4; aYAP+IR. One-way ANOVA post hoc tests. *P < 0.05.

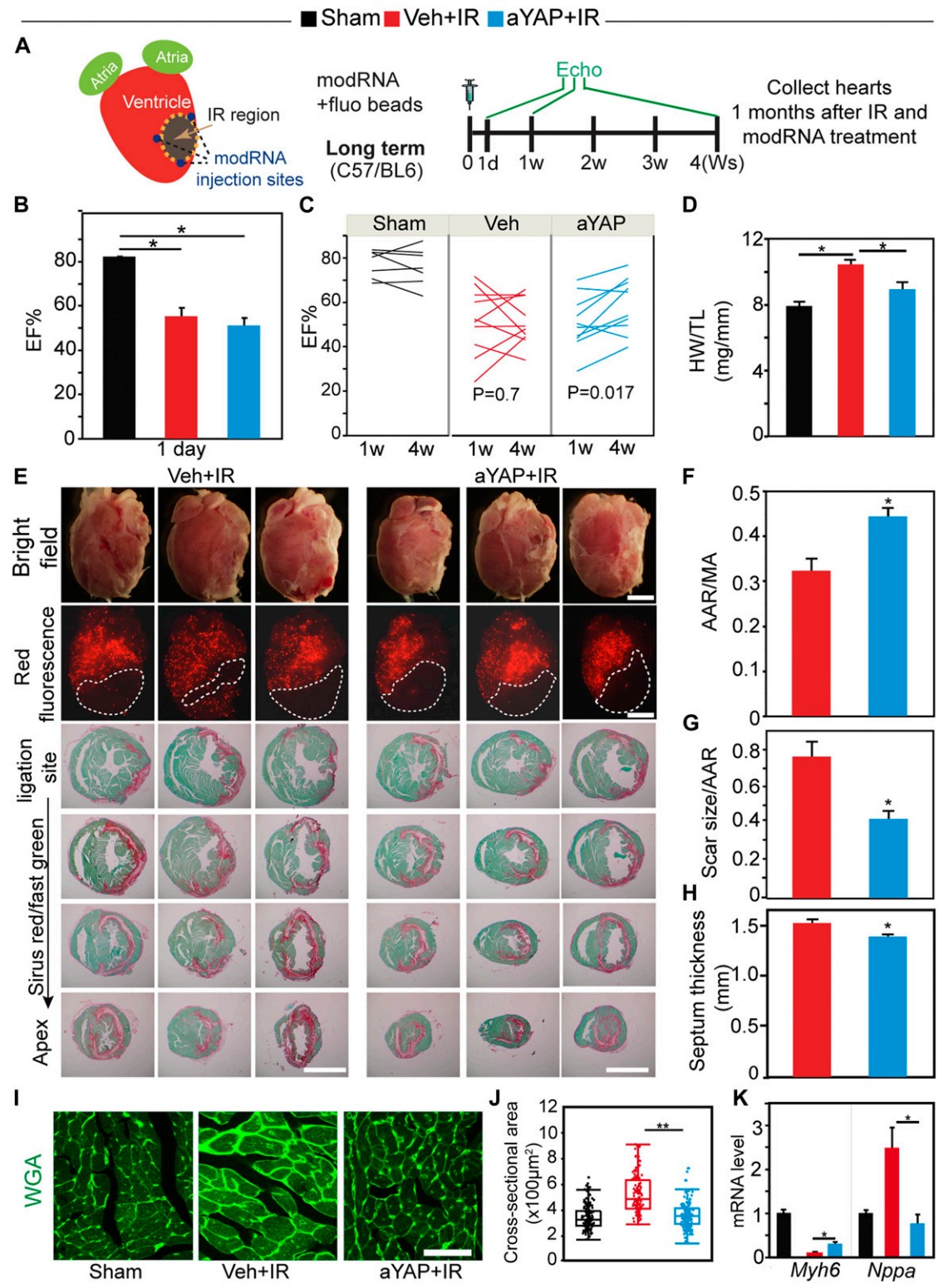

**Figure 4. aYAP modRNA improves heart function and suppresses cardiac hypertrophy.**
**(A)** Experimental design. IR and modRNA injection was performed as described in the material and methods. Red fluorescence beads were injected into the left ventricle after LAD ligation to label blood perfused ventricle. In this long term study, mouse heart function was measured by echocardiography at 1 d, 1, and 4 wk after IR. 1 mo after IR and modRNA treatment, hearts were collected for analysis. **(B)** EF% measured by echocardiography at 1 d after IR. Sham, n = 4; Veh+IR, n = 6; aYAP+IR, n = 5. **(C)** EF % measured by echocardiography. EF% was analyzed by paired t test. **(D)** Heart weight and tibia length ratio. Sham, n = 6; Veh+IR, n = 10; aYAP+IR, n = 10. One-way ANOVA post hoc tests. *P < 0.05. **(E)** Upper panel: morphology and AAR of hearts receiving different treatment. Micro red fluorescence beads were used to indicate blow flow. AAR

Together, these data suggest that aYAP modRNA treatment reduces the innate immune inflammatory response after IR.

### aYAP modRNA treatment improves heart function and suppresses hypertrophic remodeling

We designed a series of experiments to measure the effects of transient YAP activation on myocardial outcome after IR (Fig 4A). Ejection fraction (EF%), a parameter of systolic heart function, was measured by echocardiography at 1 d, 1, and 4 wk after IR and aYAP modRNA treatment. Heart rates were not significantly different between groups (Fig S4A). At 1 d after IR, both Veh+IR and aYAP+IR animals had lower systolic heart function compared with sham, and no difference was detected between Veh+IR and aYAP+IR (Fig 4B). At 1 wk after IR, the majority of mice in both Veh+IR and aYAP+IR had abnormally low EF% (lower than 60%). At 4 wk after IR, the heart function of Veh+IR animals showed a mixed pattern, with some exhibiting increased and others decreased EF% compared with 1 wk. As a result, there was no significant difference between 1 and 4 wk in the Veh+IR group (paired $t$ test: $P$ = 0.7; Fig 4C). In contrast, heart function improved in most aYAP+IR mice between 1 and 4 wk (paired $t$ test: $P$ = 0.017; Fig 4C). To clarify whether the administration of modRNA non-specifically protects the heart, we included luciferase (Luci) modRNA as a control. Similar with Veh+IR, the EF% of the Luci modRNA+IR animals did not recover 4 wk after IR (Fig S4B), suggesting that modRNA does not have non-specific cardiac protective activity.

We next assessed the effect of aYAP modRNA on hypertrophic cardiac remodeling at 4 wk after IR. Although aYAP+IR hearts had greater AAR, they appeared smaller than the Veh+IR hearts (Fig 4E and F). This was corroborated by the heart weight to tibia length ratio and heart to body weight ratio, which were both significantly lower in aYAP+IR compared with Veh+IR and was not statistically different from sham (Figs 4D and S4C). Measurement of CM cross-sectional area on histological sections further confirmed that aYAP reduced cardiac hypertrophy after IR: septum thickness of aYAP+IR was significantly lower than Veh+IR (Fig 4H). Moreover, cross-sectional area of aYAP+IR septal CMs was significantly lower than their Veh+IR counterparts, and not significantly different from sham (Fig 4I and J). Reduced hypertrophic remodeling in aYAP+IR was further supported by measurement of *Myh6* and *Nppa*, which are commonly down-regulated and up-regulated, respectively, in cardiac hypertrophy. Compared with sham, Veh+IR mice had decreased *Myh6* and increased *Nppa* expression, indicating the presence of pathological hypertrophic remodeling. Interestingly, expression of these genes was partially normalized by aYAP modRNA (Fig 4K), suggesting that transient activation of aYAP is beneficial for suppressing IR-induced hypertrophic remodeling.

Next, we evaluated the effect of aYAP modRNA on scar size after IR. Serial cardiac cross sections were stained with Sirius Red/fast green,

which stains fibrotic tissue red and myocardium green. aYAP+IR hearts had significantly smaller scar size than Veh+IR (Fig 4E and G).

Together, these data indicate that aYAP modRNA at the time of IR reduces cardiac scar size, improves cardiac function, and reduces cardiac hypertrophic remodeling.

### YAP/TEAD1 negatively regulates the expression of innate immune genes

YAP is known to regulate the innate immune response to pathogen infection (Wang et al, 2017), but whether it regulates innate immune responses in the setting of inflammation from tissue injury is not clear. YAP is a transcriptional co-activator. In the heart, the transcription factor through which YAP primarily acts is TEAD1 (von Gise et al, 2012). To test the hypothesis that YAP regulates the innate immune response through TEAD, we did gene set enrichment analysis (GSEA) (Subramanian et al, 2005) with published YAP gain-of-function (von Gise et al, 2012) and TEAD1 loss-of-function microarray data (Liu et al, 2017). 811 immunology-related genes collected in the InnateDB database (Breuer et al, 2013) were used as the GSEA gene set. In Tead1$^{cKO}$ mouse heart, the InnateDB genes were enriched in the up-regulated genes (Fig 5A). In aYAP over-expressing (OE) NRVMs, these genes were enriched in the down-regulated genes (Fig 5A). Intersection of the immunology-related genes, Tead1$^{cKO}$ differentially expressed (DE) genes, and aYAP DE genes yielded 67 immunology-related genes downstream of TEAD1 and YAP (Figs 5B and S5A). Among the top five gene ontology terms that were over-represented, 67 genes were one term related to TLR signaling and two related to cytokine signaling (Fig 5C).

Because TLR signaling regulates cytokine expression (Newton & Dixit, 2012), we hypothesized that YAP/TEAD1 regulates the expression of TLR signaling genes, which then affect the expression of cytokines. Therefore, we focused on deciphering the relationship between YAP and TLR signaling. Among the 67 genes in the intersection of the Tead1$^{cKO}$ and Yap$^{GOF}$ experiments, six TLR signaling genes were differentially regulated downstream of both YAP and TEAD1: *Tlr2*, *Tlr4*, *Cd14*, *Mapk14*, *Irak1*, and *RelA*. Among these six genes, *Tlr2*, *Tlr4*, and *Cd14* were most strongly altered by YAP activation or TEAD1 ablation (Fig 5D). With qRT-PCR, we confirmed that aYAP OE suppressed the expression of these three genes in the NRVMs (Fig 5E). We further asked whether these three genes were up-regulated in the Yap conditional knockout mouse (Yap$^{cKO}$) heart (Del Re et al, 2013). Cd14 was not DE by depletion of Yap (Fig 5F). We further extended the panel to include the six highest expressed *Tlr* genes (*Tlr1-Tlr6*) (Nishimura & Naito, 2005). In the normal mouse heart, *Tlr1* was not detected, and *Tlr2*, *Tlr3*, and *Tlr4* were more highly expressed than *Tlr5* and *Tlr6*. Compared with control, the Yap$^{cKO}$ heart had increased expression of *Tlr2* and *Tlr4*, and no change for *Tlr3*, *Tlr5*, or *Tlr6* (Fig 5G). These data suggest that YAP/TEAD1 negatively regulates the expression of a subset of innate immune genes, most notably *Tlr2* and *Tlr4*.

region without red fluorescence beads was depicted by dotted lines. Lower panel: Sirius Red and fast green staining of heart sections. Bar = 1 mm. **(F)** Ratio between AAR and left ventricle surface area. N = 5. **(G)** Scar size normalized with AAR. **(H)** Septum thickness. **(F, G, H)** N = 5. $t$ test *$P$ < 0.05. **(I)** WGA staining of heart cross sections. Bar = 50 μm. **(J)** Quantification of septum CM cross-sectional area. In each group, 150 septum CMs were randomly measured. Mann–Whitney test: **$P$ < 0.01. **(K)** qRT-PCR measurement of Mhy6 and Nppa mRNA level. N = 4. One-way ANOVA post hoc tests. *$P$ < 0.05.

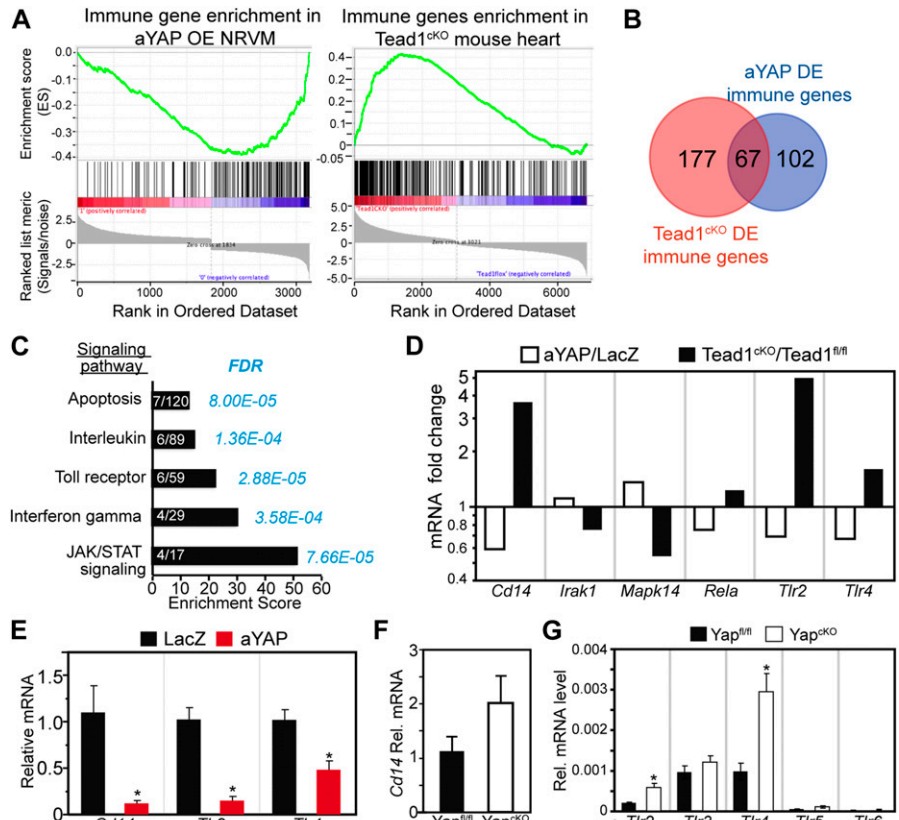

**Figure 5. YAP/TEAD1 regulates innate immune gene expression.**
**(A)** GSEA. DE (DE) genes from YAP gain-of-function or Tead1 loss-of-function data set were analyzed against an immunology gene list. YAP OE NRVMs with YAP gene overexpressed. Tead1cKO, heart-specific Tead1 knockout. Immune genes were enriched in the down-regulated genes in YAP OE NRVM and in the up-regulated genes of Tead1cKO heart. **(B)** Venn diagram of three different gene list sets: immunology genes, DE genes in the Tead1cKO data set, and DE genes in the YAP OE NRVM data set. In these three gene list sets, 67 common genes were identified. **(C)** Signaling pathway GO term analysis. The list of 67 genes identified in (B) was used for GO term signaling pathway enrichment analysis (http://geneontology.org). PANTHER overrepresentation test tool was used to analyze the enriched signaling pathways. Bar graph was plotted based on enrichment score. Gene number enriched in related pathways was labeled on each bar. Statistical significance was shown by false discovery rate value. **(D)** mRNA of Cd14 and Toll receptors and in YapcKO mouse heart. The expression values of different Tlr genes were normalized to Gapdh. RNA was collected from 1 mo old Myh6:Cre; Yapfl/fl (YapcKO) mouse heart. N = 4. *P < 0.05. **(E)** qRT-PCR measurement of Cd14, Tlr2, and Tlr4 in the NRVMs. NRVMs were treated with indicated virus for 48 h in the absence of serum. N = 4. t test: *P < 0.05. **(F, G)** mRNA of Cd14 (F) or Tlr genes (G) in 1 mo old Myh6:Cre; Yapfl/fl (YapcKO) mouse hearts. The expression values of different Tlr genes were normalized to GAPDH. N = 4. t test: *P < 0.05.

## Activation of YAP partially blunts TLR signaling in neonatal cardiomyocytes

After acute MI, innate immune signal molecules, such as DAMPs, are released from necrotic cells (Kaczmarek et al, 2013), and reperfusion of the injured myocardium brings oxidative stress to the viable cardiomyocytes (González-Montero et al, 2018). To investigate the molecular mechanisms of YAP protecting cardiomyocytes from IR stress, we tested whether YAP would suppress CM necrosis and innate immune response. We activated innate immune response in cultured CMs using LPS purified from *Escherichia coli*, a specific TLR4 agonist that is well established as a tool to study TLR4 signaling in cultured NRVMs (Hickson-Bick et al, 2006). We treated serum-starved NRVMs with both LPS and either aYAP or LacZ (control) adenovirus. The OE of aYAP was validated by Western blot (Fig S5A). Consistent with previous studies (Hickson-Bick et al, 2006), LPS treatment decreased NRVM cell viability (Fig 6A and B). This effect was blocked by aYAP OE, which increased viability to at least control levels (Fig 6A and B).

We further asked whether aYAP improved NRVM cell viability by decreasing necrosis or apoptosis. We used propidium iodide (PI), which labels cells that have lost cell membrane integrity, and Apopxin Green, which labels cells with exposed phosphatidylserine, to mark necrotic and apoptotic CMs, respectively. Compared with LacZ, LacZ+LPS had significantly more PI+ cells and similar Apopxin+ cells (Fig 6C–F), indicating that LPS increases CMs necrosis but not apoptosis. In aYAP+LPS, the PI+ cells were reduced to below baseline (LacZ) levels (Fig 6C and D), suggesting that activation of

YAP blocks LPS-induced necrosis. LPS+YAP also had the lowest percentage of Apopxin+ cells, but this did not reach statistical significance (Fig 6E and F).

The Drosophila homolog of YAP controls the innate immune response by activating expression of Cactus, whose mammalian homologs, Nfkbia and Nfkbib, are key inhibitors of the pro-inflammatory transcription factor NF-κB. Therefore, we measured the effect of aYAP OE on *Nfkbia* and *Nfkbib*. As internal controls to confirm the expected effects of aYAP on gene expression, we also measured the expression of well-characterized YAP target genes *Ctgf*, *Cyr61*, and *Pik3cb* (Yu et al, 2012; Lin et al, 2015). LPS did not significantly affect the expression of these YAP target genes because their expression was similar between LacZ+LPS and LacZ. In aYAP+LPS, *Ctgf*, and *Pik3cb* were significantly up-regulated compared with LacZ+LPS or LacZ only, but the expression of *Cyr61* was not significantly changed (Fig 6G). Comparison of LacZ+LPS to LacZ showed that both *Nfkbia* and *Nfkbib* were up-regulated by LPS (Fig 6H). In the presence of LPS treatment, OE of aYAP returned the expression of *Nfkbia* and *Nfkbib* to baseline levels (Fig 6H). This suggests that YAP inhibition of the innate immune response in the heart does not proceed through activation of NF-κB inhibitors Nfkbia and Nfkbib as it does in Drosophila.

To further investigate the mechanisms by which YAP reduces the cardiac innate immune response, we measured the effect of LPS-induced innate immune activation on the expression of *Tlr2*, *Tlr4*, and *Cd14* in the NRVM system. Consistent with studies carried out in macrophages (Matsuguchi et al, 2000), LPS increased expression of

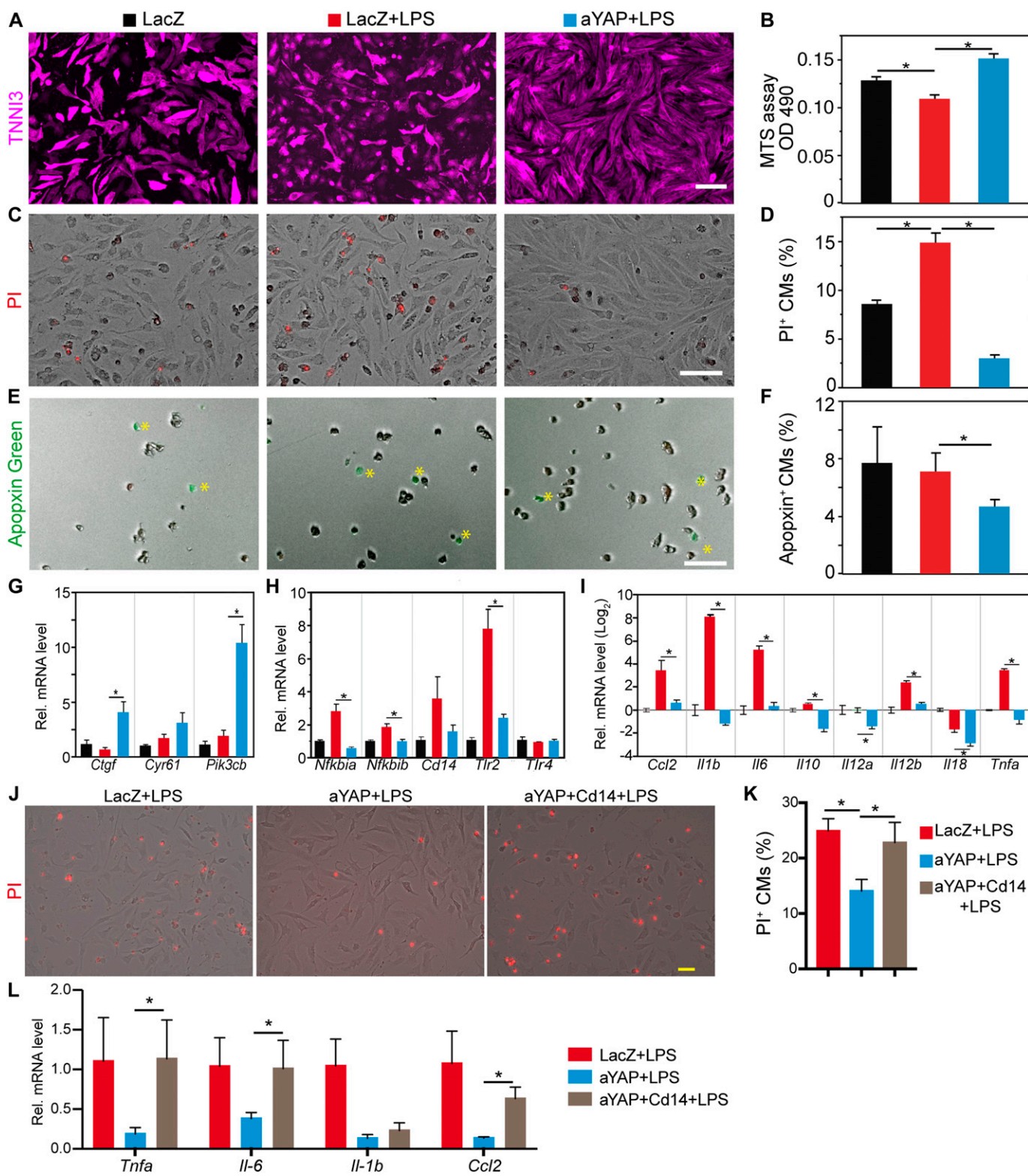

**Figure 6. aYAP suppresses TLR4 signaling in vitro.**
In the absence of serum, NRVMs were first treated with the indicated virus for 12 h, then 1 µg/ml LPS was added to activate TLR4 signaling. 24 h after LPS stimulation, NRVMs were collected for gene expression analysis or cell death studies. **(A)** Cell morphology after LPS and indicated adenovirus treatment. Cardiac troponin I (TNNI3) was used to label CMs. Bar = 100 µm. **(B)** Cell viability assay. Cells were treated with MTS solutions and OD 490 was measured according to manufacturer's protocol (Promega). N = 6. **(C, D)** NRVM necrosis analysis. **(C)** Representative images of PI stained NRVMs. Bar = 100 µm. **(D)** Quantification of PI-positive CMs. N = 4. **(E, F)** NRVM apoptosis analysis. **(E)** CMs stained with Apopxin Green. Bar = 100 µm. **(F)** Quantification of Apopxin-positive CMs. N = 4. **(G, H, I)** NRVM gene expression analysis. **(G)** mRNA level of

*Tlr2* and *Cd14* (Fig 6H). The strong up-regulation of *Tlr2* by LPS was blocked by aYAP (Fig 6H). *Cd14* activation was also reduced by aYAP, although the effect did not reach statistical significance (Fig 6H). Unlike *Tlr2*, LPS did not induce *Tlr4*. In the baseline condition, YAP suppressed the expression of *Tlr4* (Fig 5E); however, aYAP did not affect *Tlr4* expression in the presence of LPS (Figs 6H and S5B), suggesting that YAP regulation of *Tlr4* is context-dependent.

TLR signaling activates expression of cytokines/chemokines. We measured the expression of several cytokine and chemokine transcripts *Ccl2*, *Il-1b*, *Il-6*, *Il-10*, *Il-12a*, *Il-12b*, *Il-18*, and *Tnfa*. LPS robustly increased the expression of *Ccl2*, *Il-1b*, *Il-6*, *Il-10*, *Il-12b*, and *Tnfa*, decreased *Il-18*, and had no effect on *Il-12a* (Fig 6I). Expression of all of these transcripts was significantly lower in aYAP+LPS compared with LacZ+LPS (Fig 6I). Together, these data indicate that LPS induced a subset of innate immune response genes in NRVMs, and this effect was inhibited by aYAP.

CD14 is a crucial co-factor mediating TLRs signaling (Lee et al, 2012). To further test whether YAP reduced CM innate immune response by blunting TLR signaling, we generated an adenovirus OE CD14 (Fig S5F). Although aYAP significantly suppressed LPS-induced CM necrosis, this effect was abolished in the presence of excessive CD14 (Fig 6J and K). Meanwhile, compared with aYAP+LPS group, the expression of *Tnfa*, *Il-6*, and *Ccl2* were all increased in the LPS+YAP+CD14 group (Fig 6L). These results suggest that YAP blunts CM necrosis and innate immune response by inhibiting TLR signaling pathway.

In acute MI, oxidative stress plays a detrimental role by causing tissue necrosis and reperfusion injury (Giordano, 2005). In vitro, increased oxidative stress can be modeled by treatment with $H_2O_2$. YAP has been shown to suppress $H_2O_2$-induced cardiomyocyte necrosis (Del Re et al, 2013), but it is not clear whether YAP also suppresses $H_2O_2$-induced cardiomyocyte innate immune responses (Zhang et al, 2012). We treated NRVMs with $H_2O_2$ and tested the effects of YAP. Consistent with the published data (Del Re et al, 2013), YAP activation decreased $H_2O_2$-induced CM necrosis (Fig S5C and D). $H_2O_2$ decreased the expression of *Il-12a* and *Tlr2* but had no effect on *Tlr4* (Fig S5E), and activation of YAP did not affect the expression of these three genes in the presence of $H_2O_2$. These data suggest that YAP protects the CMs from oxidative damage independent of TLR signaling.

### Activation of YAP reduces adult cardiomyocyte necrosis

Neonatal CMs are different from adult cardiomyocytes in many respects (Scuderi 2 Butcher, 2017), and they may have different responses to environmental stress. We, therefore, repeated our studies in cultured adult CMs to assess whether YAP would protect adult cardiomyocytes from LPS and $H_2O_2$-induced necrosis. Same with NRVMs, activation of YAP significantly decreased LPS- or $H_2O_2$-induced adult CMs necrosis (Fig 7A–D). In the adult CMs, LPS did not affect the expression of *IL-12a* and *IL-6*, but increased the expression of *Ccl2* and *Il-10*. In the presence of LPS, activation of YAP decreased the expression of *IL-12a* and had no effects on the other three genes (Fig 7E). In NRVMs, LPS induced the expression of *Tlr2*

but not *Tlr4* (Fig 6H). Different from NRVMs, LPS increased the expression of both *Tlr2* and *Tlr4* in the adult CMs (Fig 7F). Similar to NRVMs, in the adult CMs, LPS induction of *Tlr2* but not *Tlr4* was suppressed by YAP (Fig 7F).

In the adult CMs, $H_2O_2$ treatment substantially increased the expression of *Il-12a*, *Ccl2*, *Il-6*, and *Il-10* (Fig 7E). In the presence of $H_2O_2$, activation of YAP did not change the expression of *Il-12a* and *Il-6* but increased the expression of *Ccl2* and *Il-10*. Same with NRVMs, $H_2O_2$ did not affect the expression of *Tlr2* and *Tlr4*. In the presence of $H_2O_2$, activation of YAP did not affect the expression of *Tlr2* and *Tlr4* (Fig 7F).

We isolated cardiomyocytes from the Veh+IR and aYAP+IR hearts, and tested the expression of *Tlr2*, *Tlr4*, *Il-6*, *Il-10*, and *Ccl2*. IR stress induced the expression of these five genes, aYAP modRNA treatment decreased *Tlr2*, increased *Il-10*, and had no effect on the other three genes (Fig 7G and H).

Together, these data indicate that that aYAP improves CM survival through TLR-dependent and TLR-independent pathways and that YAP suppresses CM innate immune by suppressing the TLR signaling pathway (Fig 7I).

## Discussion

After a heart attack, timely relief of coronary artery occlusion is essential for salvaging affected CMs. However, reperfusion itself also causes injury to the myocardium (Prasad et al, 2009). In this study, we showed that transient activation of YAP using aYAP modRNA improved cardiac systolic function and suppressed cardiac hypertrophic remodeling in a mouse IR model by decreasing CM necrosis, attenuating cardiac inflammation, and reducing scar formation.

In vitro-transcribed mRNA has been used for gene delivery and vaccination in an array of preclinical studies, but its instability and immunogenicity largely limited its application (Sahin et al, 2014). Technologies have been developed to produce modified mRNAs (modRNA) that are more stable and less immunogenic than the unmodified mRNA (Karikó et al, 2005). modRNA has been shown to successfully drive target genes expression in different animal disease models. For example, intratracheal delivery of Foxp3 modRNA to the lung protects against asthma (Mays et al, 2013), and intramyocardial delivery of VEGF modRNA promotes vascular regeneration after MI (Zangi et al, 2013). The VEGF modRNA was recently tested in a swine MI model (Carlsson et al, 2018) and type II diabetes patients (Gan et al, 2019), which highlighted the safety and efficacy of modRNA as a gene delivery tool. Since the discovery of apoptosis, many signaling pathways regulating CM death have been characterized (Chiong et al, 2011). Nevertheless, tools to temporally manipulate these signaling pathways to salvage the myocardium are limited. In this study, we used aYAP modRNA to target the short heart repair window after IR, and the results suggest that transient activation of the transcriptional factor or co-factors with modRNA is a good strategy to reduce cardiomyocyte loss and help cardiac repair. Therefore, our study opened a new avenue

known YAP target genes. **(H)** mRNA level of crucial innate immune genes regulated by YAP. **(I)** mRNA level of selected cytokine/chemokine genes. N = 4. **(J, K)** NRVM necrosis analysis. **(J)** Representative images of PI stained NRVMs. Bar = 50 μm. **(K)** Quantification of PI-positive CMs. N = 4. **(L)** NRVM gene expression analysis. N = 4. **(B, D, F, G, H, I, K, L)** One-Way ANOVA Post Hoc Tests. *$P < 0.05$.

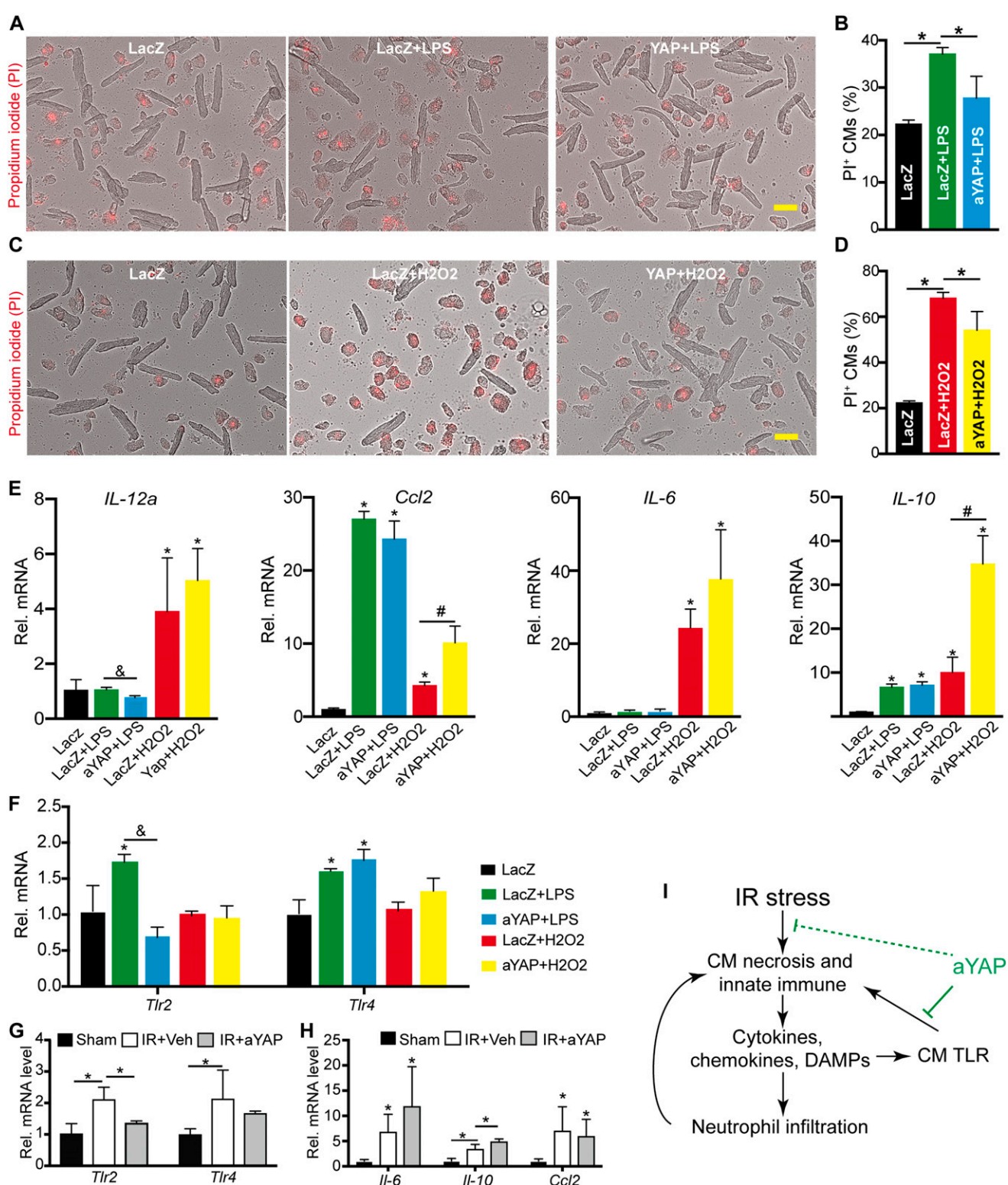

**Figure 7. aYAP improves adult cardiomyocyte survival in vitro.**
**(A, B, C, D)** Dissociated adult mouse cardiomyocytes (CMs) were treated with indicated conditions (see details in Materials and Methods section). PI was used to label necrotic cells. **(A, C)** Representative images of ACMs. Bar = 50 µm. **(B, D)** Quantification of adult CM death rate. **(B, D)** n = 4. One-way ANOVA post hoc tests. *$P < 0.05$. **(E, F, G, H)** qRT-PCR measurement of gene expression. Total RNA was collected from adult CMs treated with indicated conditions. The expression of different genes was normalized to GAPDH. Gene expression values were analyzed by One-way ANOVA post hoc tests. *$P < 0.05$, gene expression values were compared between experimental groups and LacZ control group. &, *$P < 0.05$, values were compared between LPS+LacZ and LPS+aYAP. #$P < 0.05$, values were compared between LacZ+$H_2O_2$ and aYAP+$H_2O_2$. N = 3. **(I)** Schematic summary of the current study. Blunt ended green lines indicate processes suppressed by aYAP.

for translating the current CM death-related knowledge into therapeutic use.

Although aYAP modRNA is transiently expressed in the myocardium, our data show that aYAP modRNA has sustained beneficial effects on myocardial recovery after IR injury. Because of the post-mitotic nature of adult cardiomyocytes (CMs), CMs lost during IR cannot be replenished. As a result, to provide enough pumping power, the remaining CMs in the injured heart undergo pathological hypertrophic remodeling. In acute MI patients, infarct size is linearly anti-correlated with the amount of salvaged myocardium, and it is the most important predictor of adverse ventricular remodeling (Orn et al, 2007). Consistent with the observation that transient activation of YAP reduces CM death and myocardial scar size, the pathological hypertrophic remodeling of the YAP modRNA treated mice is suppressed after IR injury (Fig 4). It is possible that transient activation of YAP after IR helps to salvage the injured myocardium, which in the long run attenuates the pathological hypertrophic remodeling.

After IR stress, many CMs undergo necrosis and release DAMPs, which attract neutrophils and macrophages to the injury site (Frangogiannis, 2012). In large animal IR studies, higher neutrophil infiltration is associated with larger infarct size (Litt et al, 1989), suggesting that neutrophil influx after reperfusion is detrimental to the myocardium. In our study, aYAP modRNA significantly reduced CM death, and this was associated with reduced circulating neutrophils and lower myocardial neutrophil infiltration. The reason for aYAP modRNA reducing neutrophils in both the myocardium and the peripheral blood is not clear. One possible explanation is that aYAP modRNA has accumulative effects on reducing DAMPs and cytokines release, and therefore, attenuates neutrophil infiltration and bone marrow to peripheral blood mobilization in a time-dependent manner. This hypothesis is supported by the modRNA translation kinetics data, which shows that the translation starts as soon as 3 h after transfection and takes 1 d to reach its peak (Zangi et al, 2013). In addition, aYAP modRNA did not reduce myocardial neutrophils on the first day after IR, further suggesting that aYAP modRNA may not directly suppress the neutrophils infiltration. Nevertheless, more data are required to address how aYAP modRNA affects neutrophil infiltration and mobilization.

We used two in vitro culture models to interrogate the protective roles of YAP: NRVMs and adult CMs. In both models, activation of YAP suppresses LPS-induced CM necrosis and *Tlr2* expression; however, YAP does not affect $H_2O_2$ suppression of *Tlr2*. These data suggest that YAP protects CM through TLR dependent and independent pathways. The responses of NRVMs and adult CMs to $H_2O_2$ are largely different. For example, adult CMs are more vulnerable to $H_2O_2$ treatment than NRVMs, and adult CMs express more cytokine genes (such as Il-12a) (Heufler et al, 1996) than the NRVMs under $H_2O_2$ stress. In addition, in the presence of $H_2O_2$, YAP suppresses *Il-10* in the NRVMs but robustly increases its expression in the adult CMs. Because we used adult animals for IR studies, the data from cultured adult CMs are more reliable for interpreting the in vivo observations. Therefore, our data suggest that YAP protects the CM from necrosis, and that YAP also attenuates inflammation by reducing inflammatory cytokine (such as Il-12a) and increasing anti-inflammatory cytokine (such as Il-10) (Couper et al, 2008) genes expression.

Recent studies indicate that Hippo-YAP pathway regulates innate immune responses in both Drosophila and mammals (Liu et al, 2016; Wang et al, 2017). In this study, we specifically addressed the roles of YAP in the regulation of CM innate immune. NRVMs and adult CMs have similar sensitivity to LPS treatment. In both NRVMs and adult CMs, YAP decreases LPS-induced CM necrosis and attenuates LPS induction of *Il-12a* and *Tlr2*, suggesting that YAP indeed suppresses TLR signaling in the CMs. In Drosophila, YAP activates inhibitors of NF-kB, a key inflammatory transcription factor; however, our data indicate that YAP's mechanism of action is different in the heart because it suppressed, rather than activated, these inhibitors. A large body of literature shows that TLR signaling is a crucial driver of the cardiac innate immune response (Knowlton, 2017). We show that YAP/TEAD1 suppresses the expression of many immunology-related genes in the heart, most notably *Cd14*, and *Tlr2*, core components of TLR signaling (Knowlton, 2017). Our results are different from a recent study, which showed that deletion of Yap in the macrophages did not affect the expression of *Tlr4* (Zhou et al, 2019). Together, these data suggest that YAP might have different roles in CMs and macrophages, and that the regulation of *Tlr4* expression might be cell context-dependent.

We also observed that YAP/TEAD1 reduced *Tlr4* expression in some but not all contexts. Furthermore, YAP protected NRVMs from LPS, a specific activator of TLR4, by blocking LPS-induced cell death and activation of multiple cytokines and chemokines. Protection from LPS was not through decreased *Tlr4* expression. Our data point to two likely mechanisms. First, TLR4 signaling requires both TLR2 and CD14 (Good et al, 2012; Conti et al, 2014), both of which were down-regulated by aYAP. Second, Pik3cb, a catalytic subunit of phosphoinositide 3-kinase, negatively regulated TLR-mediated responses in macrophages (Hazeki et al, 2006). We previously identified *Pik3cb* as a direct target of YAP in CMs (Lin et al, 2015), and YAP strongly increased CM *Pik3cb* expression in the context of LPS stimulation. Third, activation of CD14 in NRVMs abolished YAP's protective effects against LPS. These data suggest that YAP blunts LPS-TLR4 signaling by affecting the expression of *Cd14*, activating phosphoinositide 3-kinase, or both. Our data suggest that although YAP regulates CM innate immune system by inhibiting TLR4 signaling pathway, it is possible that YAP may also suppress cardiomyocyte innate immune response through other mechanisms, such as by suppressing IRF3 (Wang et al, 2017). Although intriguing, testing this hypothesis is out of the scope of this work and could be pursued in future studies.

One limitation of this study is that the aYAP modRNA was expressed in multiple cell types, such as CMs and non-CMs. Although we focused on its effect in CMs, aYAP modRNA may have also influenced the function of other cell types. A second limitation of the study is that our delivery route has limited translational relevance. Here, we focused on the biological properties of the aYAP modRNA rather than development of a more translational delivery route. Future development efforts will need to improve modRNA delivery to broaden its potential translational applications.

In summary, this study demonstrates that transient activation of YAP by aYAP modRNA has sustained beneficial effects in a mouse IR model. In addition to the now established role of YAP to stimulate

cardiac regeneration, we show that the beneficial activity of YAP is also mediated through inhibition of necrosis and innate immune signaling.

# Materials and Methods

Please see Supplemental Methods section of Supplemental Data 1 for details.

### Experimental animals

All animal procedures were approved by the Boston Children's Hospital Animal Care and Use Committee. C57BL/6J mice aged 6–8 wk were obtained from Jackson Labs. IR surgery was performed blinded to treatment group.

### Active YAP modified RNA (aYAP modRNA)

Triple FLAG epitope-tagged aYAP modRNA was in vitro transcribed and purified by HPLC in the RNA Therapeutics Core, Houston Methodist Research Institute. Before injection, aYAP modRNA was diluted to 0.9% NaCl at a concentration of 0.67 $\mu g/\mu l$.

### Echocardiography

Echocardiography was performed at 1, 7 d, and 4 wk after IR, using a VisualSonics Vevo 2100 equipped with VevoStrain software. Data were acquired from awake mice. Acquisition and measurements were performed by investigators blinded to treatment group. Mice with EF% higher than 75% on the first day after IR were excluded from the following studies.

### Blood cell tests

Blood samples from affected mice were collected into heparin-coated tubes and run on a HEMAVET 950FS (Drew Scientific) auto blood analyzer. The numbers of white blood cells, red blood cells, neutrophils, and lymphocytes were automatically analyzed.

### ELISA assay

Serum samples were collected from mice euthanized at 1 and 2 d post-IR. Mouse cardiac troponin T (cTnT) concentration was measured by ELISA (MyBioSource).

### Gene expression

Western blotting was performed using specific antibodies (Table S1). Total RNA was isolated using TRIzol. For qRT-PCR, RNA was reverse transcribed (Superscript III) and specific transcripts were measured using SYBR Green chemistry and normalized to GAPDH. Primer sequences are provided in Table S2.

### Statistics

Values are expressed as mean ± SEM. $t$ test or ANOVA with Tukey's honestly significant difference post hoc test was used to test for statistical significance involving two or more than two groups, respectively.

# Supplementary Information

# Acknowledgements

We thank Alexander Von Gise from the University of Hannover for critically reviewing the manuscript. Sources of Funding: J Chen was supported by a Scholarship Fund from China Scholarship Council. Z Lin was supported by American Heart Association Scientist Development Grant 15SDG25590001, Progenitor Cell Biology Consortium "JUMP START" AWARDS, 5U01HL099997-07, and National Institutes of Health HL138454-01. WT Pu was supported by National Institutes of Health R01 HL116461.

## Author Contributions

J Chen: investigation.
Q Ma: methodology.
JS King: investigation and writing—editing.
Y Sun: investigation.
B Xu: methodology.
X Zhang: methodology.
S Zohrabian: investigation.
H Guo: investigation.
W Cai: investigation.
G Li: investigation.
I Bruno: resources.
JP Cooke: resources.
C Wang: resources.
M Kontaridis: resources.
D-Z Wang: resources.
H Luo: conceptualization and resources.
WT Pu: conceptualization, resources, supervision, funding acquisition, project administration, and writing—review and editing.
Z Lin: conceptualization, resources, data curation, formal analysis, supervision, funding acquisition, validation, investigation, methodology, project administration, and writing—original draft, review, and editing.

## Conflict of Interest Statement

The authors declare that they have no conflict of interest.

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
