## [Reviewer comments · Life Science Alliance]

Life Science Alliance

aYAP modRNA reduces cardiac inflammation and hypertrophy in a murine Ischemia-reperfusion model

Jinmiao Chen, Qing Ma, Justin King, Yan Sun, Bing Xu, Xiaoyu Zhang, Sylvia Zohrabian, Haipeng Guo, Wenqing Cai, Gavin Li, Ivone Bruno, John Cooke, Chunsheng Wang, Maria Kontaridis, Da-Zhi Wang, Hongbo Luo, William Pu, and Zhiqiang Lin

DOI: <https://doi.org/10.26508/lsa.201900424>

Corresponding author(s): Zhiqiang Lin, Masonic Medical Research Institute

Review Timeline:

Submission Date:	2019-05-10
Editorial Decision:	2019-06-27
Revision Received:	2019-08-10
Editorial Decision:	2019-08-22
Revision Received:	2019-11-13
Editorial Decision:	2019-12-03
Revision Received:	2019-12-04
Accepted:	2019-12-05

Scientific Editor: Andrea Leibfried

Transaction Report:

June 27, 2019

Re: Life Science Alliance manuscript #LSA-2019-00424-T

Dr. Zhiqiang Lin
Masonic Medical Research Institutue
2150 Bleecker Street
Utica, NY 13501

Dear Dr. Lin,

Thank you for submitting your manuscript entitled "aYAP modRNA treatment reduces cardiac inflammation and hypertrophic remodeling in a murine Ischemia-reperfusion model" to Life Science Alliance. Please excuse the delay in getting back to you, the reviewing process took a little longer than usual in this case. Your manuscript has now been assessed by expert reviewers, whose comments are appended to this letter.

As you will see, the reviewers appreciate your data and see value in your findings. They provide constructive input on how to further strengthen your work to allow publication here. We would thus like to invite you to submit a revised version to us, addressing the concerns raised by the two reviewers. Importantly, please provide more support for the suggested link of the protective effect of transient Yap activation to TLR signaling and innate immunity. Addressing all concerns of the reviewers seems rather straightforward, but please get in touch in case you would like to discuss individual points further.

Thank you for this interesting contribution to Life Science Alliance. We are looking forward to receiving your revised manuscript.

Sincerely,

B. MANUSCRIPT ORGANIZATION AND FORMATTING:

Reviewer #1 (Comments to the Authors (Required)):

In this study, the authors transiently activated YAP in IR stressed myocardium with modified mRNA encoding a constitutively active form of YAP (aYAP modRNA). Expression of active YAP modRNA protected the myocardium by reducing cardiomyocyte (CM) necrosis and neutrophil infiltration.

Four weeks after IR, YAP modRNA treated mice had better heart function, reduced scar size and less hypertrophic remodeling. Data presented in the paper suggest that YAP protects the myocardium by reducing oxidative stress-induced necrosis and suppressing Toll-like receptor innate immunity signaling. Overall this is an interesting study that can be improved by addressing comments:

Major Comments

1) "In transduced CMs, aYAP was detected in both cytoplasm and nucleus (Fig. 1F). Other cell types, such as endothelial cells, were also transduced"

The authors note that the Mod RNA is transduced into all cell types - does this have an effect in all cell types or just CMs?

2)"These data suggest that aYAP modRNA reduces the infiltration of neutrophils two days after IR".

Please discuss this observation in more depth in the discussion section

3)"Compared to sham, expression of Myh6 and Nppa were down- and up- regulated in Veh+IR. However, expression of these genes was partially normalized by aYAP"

This statement is a little confusing please rephrase and discuss in more depth

4)"These data suggest that YAP/TEAD1 negatively regulates the expression of a subset of innate immune genes, most notably Tlr2 and Tlr4."

This interesting observation should be discussed in more depth as this connection between Yap and the innate immune response has been described in other systems.

5)"Together these data indicate that LPS induced a subset of innate immune response genes in NRVMs, and this effect was inhibited by aYAP."

Please discuss how this might work as Yap is generally considered to be a transcriptional activator

Reviewer #2 (Comments to the Authors (Required)):

Chen et al describes a pro regenerative/protective role for aYAP (S127A) in ischemia- reperfusion (IR) injury of the heart. Chen et al use a transient mode of delivery, modRNA, that was injected i.m. around the infarct site. aYAP increased CM survival both in vivo and in vitro, resulting in decreased infarct size, and decreased inflammation (reduced infiltration of neutrophils and macrophages). Furthermore, they show CM cell cycle re-entry/ proliferation. Finally, aYAP expression resulted in improved cardiac function and reduced cardiac remodeling, implicating cardiac regeneration. On the mechanistic aspect, Chen et al report that YAP/TEAD regulates innate immune gene expression, and specifically TLR signaling.

Yap activation is an established player in cardiac regeneration, mainly through its ability to promote

adult CM proliferation. Therefore, some aspects of this study lack novelty. The transient nature of aYap delivery by modRNA is of high interest, as continuous mitotic signaling is deleterious to the heart, specifically considering that aYAP is a known oncogene. The role of aYap on CM survival and reduced immune cells' infiltration is of interest to the field. CM survival was demonstrated convincingly in more than one system, and it seems that both necrotic and apoptotic processes are affected.

Major issues:

1. TLR signaling suppression leads to improved CM survival: While the authors demonstrate Yap can regulate Tlr2, cd14 and pik3cb, and thereby influence Tlr4 signaling. They do not provide any direct evidence for TLR-dependent CM death. This could be a result of other Yap downstream effectors, mentioned in the discussion section. To resolve this issue, the authors could perform an over expression experiment, in vitro or in vivo, in which aYAP introduced to CM/ hearts over expressing Tlr2/ cd14/ both (i.e. by AAV vectors), or KO/ KD of pik3cb. Alternatively a genetic experiment with activation of TLR signaling in CMs, in the presence of aYAP, which should prevent aYAP effect.

Minor/ technical issues:

1. Introduction add more relevant references for Hippo/ Yap involvement in cardiac regeneration.
2. Figure S2D: images look similar, if you count only the Edu+ CMs. Suggest replacing the images, and/or doing this experiment at 7 days post injury with more EdU injections.
3. Figure 3: the legend and figures need some corrections. It's not trivial understanding what's presented. For example, adding time points (D1/ D2) to A-F would be helpful.
4. Figure 3: analyzing the same heart by FACS and histology is not clear. Variability of sectioning can lead to addition or missing of a highly inflamed region. It might be better to use different hearts for the different experiments.
5. The conclusion from the data shown in Figure 3 and Figure S3 is that aYAP reduces innate immune cells infiltration (mainly neutrophils). The way it's written now, it implies that aYAP may affect the neutrophil directly, while the simpler conclusion (shown in Fig. 7I) is that reduced infarct size leads to reduced infiltration, shown by the similar neutrophil density in the infarct of both animal groups. To rule out an initial effect of aYAP on neutrophils, as the modRNA delivery is nonspecific, it would be interesting to look at neutrophil dynamics. Neutrophils are known to be the first cells to infiltrate the injury site.
6. Figure 4B, C: the EF values of the sham seems too high. The variation in the injury is very low. What was the sample size for this experiment? Similar to D? Please state it.
7. Figure 5C: please specify the GO annotation tool used to do this. GSEA? If so, which module?
8. Figure 5E: Were the cells treated with the adenovirus as shown in Fig S5? The authors don't refer to it in the legend or text, only at Fig. S5 and 6. Also, the data in Fig 5E and Fig 6H seems contradictory with respect to Tlr4.
9. Page 11, first paragraph: when addressing the WB verifying aYAP expression, it should be "Fig. S5B" and not "Fig S5A".

Summary:

The novelty of this study is mainly the protective effect of transient Yap activation, and the ability to connect this to TLR signaling and thereby innate immunity. The second claim is still speculative, as it needs a direct evidence. Other technical issues are minor. The transient nature of this delivery system makes it interesting for translational purposes.

Reviewer #1 (Comments to the Authors (Required)):

In this study, the authors transiently activated YAP in IR stressed myocardium with modified mRNA encoding a constitutively active form of YAP (aYAP modRNA). Expression of active YAP modRNA protected the myocardium by reducing cardiomyocyte (CM) necrosis and neutrophil infiltration.

Four weeks after IR, YAP modRNA treated mice had better heart function, reduced scar size and less hypertrophic remodeling. Data presented in the paper suggest that YAP protects the myocardium by reducing oxidative stress-induced necrosis and suppressing Toll-like receptor innate immunity signaling. Overall this is an interesting study that can be improved by addressing comments:

We thank the reviewer for providing these nice comments. The focus of this study is to test the efficacy of Yap modRNA in the treatment of IR injury. Follow-up studies are being carried on to test the detail molecular mechanisms about how Yap modRNA reduces cardiac inflammatory after IR stress.

Major Comments

1) "In transduced CMs, aYAP was detected in both cytoplasm and nucleus (Fig. 1F). Other cell types, such as endothelial cells, were also transduced"

The authors note that the Mod RNA is transduced into all cell types - does this have an effect in all cell types or just CMs?

Response: We believe that the modRNA has responses to other cell types, such as endothelial cells. We used CM survival as a readout of Yap modRNA treatment, because either directly or indirectly salvaging the CMs is the final goal of reducing reperfusion injury. Due to the limit of the current modRNA delivery method, we are unable to address the function of YAP modRNA in different cell types in this study. We have two sentences in our discussion part on page 18 to point out the limitation of this study: "One limitation of this study is that the aYAP modRNA was expressed in multiple cell types, such as CMs and non-CMs. Although we focused on its effect in CMs, aYAP modRNA may have also influenced the function of other cell types."

2)"These data suggest that aYAP modRNA reduces the infiltration of neutrophils two days after IR".

Please discuss this observation in more depth in the discussion section

Response: We put several sentences in the discussion section on page 16 to explain this observation.

3)"Compared to sham, expression of Myh6 and Nppa were down- and up- regulated in Veh+IR. However, expression of these genes was partially normalized by aYAP"

This statement is a little confusing please rephrase and discuss in more depth

Response: We rephrased this sentence as " Compared to sham, Veh+IR mice had decreased Myh6 and increased Nppa expression, indicating the presence of pathological hypertrophic remodeling. Interestingly, expression of these genes was partially normalized by aYAP modRNA, suggesting that transiently activation of aYAP is beneficial for suppressing IR-induced hypertrophic remodeling."

4)"These data suggest that YAP/TEAD1 negatively regulates the expression of a subset of innate immune genes, most notably Tlr2 and Tlr4."

This interesting observation should be discussed in more depth as this connection between Yap and the innate immune response has been described in other systems.

Response: We put two sentences to the discussion section on page 17: "Our results are different from a recent study, which showed that deletion of Yap in the macrophages did not affect the expression of Tlr4 (Zhou et al., 2019). These data suggest that YAP might have different roles in CMs and macrophages, and that the regulation of *Tlr4* expression might be cell context dependent."

5)"Together these data indicate that LPS induced a subset of innate immune response genes in NRVMs, and this effect was inhibited by aYAP."

Please discuss how this might work as Yap is generally considered to be a transcriptional activator

Response: YAP has binary roles in the regulation of target genes expression, and it can serve as both activator and suppressor (Kim, Kim, Johnson, & Lim, 2015) depending on the biological context. For example, it is known that YAP maintains human embryonic stem cell pluripotency by directly suppressing the expression of *WNT3* (Estarás, Hsu, Huang, & Jones, 2017). In this study, our data suggest that YAP negatively regulates the expression of some innate immune genes in the CMs.

Reviewer #2 (Comments to the Authors (Required)):

Chen et al describes a pro regenerative/protective role for aYAP (S127A) in ischemia-reperfusion (IR) injury of the heart. Chen et al use a transient mode of delivery, modRNA, that was injected i.m. around the infarct site. aYAP increased CM survival both in vivo and in vitro, resulting in decreased infarct size, and decreased inflammation (reduced infiltration of neutrophils and macrophages). Furthermore, they show CM cell cycle re-entry/ proliferation. Finally, aYAP expression resulted in improved cardiac function and reduced cardiac remodeling, implicating cardiac regeneration. On the mechanistic aspect, Chen et al report that YAP/TEAD regulates innate immune gene expression, and specifically TLR signaling. Yap activation is an established player in cardiac regeneration, mainly through its ability to promote adult CM proliferation. Therefore, some aspects of this study lack novelty. The transient nature of aYap delivery by modRNA is of high interest, as continuous mitotic signaling is deleterious to the heart, specifically considering that aYAP is a known oncogene. The role of aYap on CM survival and reduced immune cells' infiltration is of interest to the field. CM survival was demonstrated convincingly in more than one system, and it seems

that both necrotic and apoptotic processes are affected.

Major issues:

1. TLR signaling suppression leads to improved CM survival: While the authors demonstrate Yap can regulate Tlr2, cd14 and pik3cb, and thereby influence Tlr4 signaling. They do not provide any direct evidence for TLR-dependent CM death. This could be a result of other Yap downstream effectors, mentioned in the discussion section. To resolve this issue, the authors could perform an over expression experiment, in vitro or in vivo, in which aYAP introduced to CM/ hearts over expressing Tlr2/ cd14/ both (i.e. by AAV vectors), or KO/ KD of pik3cb. Alternatively a genetic experiment with activation of TLR signaling in CMs, in the presence of aYAP, which should prevent aYAP effect.

Response: We totally agree with the reviewers comments. In this study, we focused on studying the efficacy of aYAP modRNA, to see whether it would reduce IR-induced inflammation and injury. YAP negatively regulates the expression of Tlr2 and Tlr4, but YAP-TLR axis is not necessary the major molecular mechanism of preventing CM necrosis. Therefore, during our study, we did not claim YAP suppress CM necrosis by suppressing Tlr4 signaling pathway. Rather, in the discussion section, we claimed that "...YAP blunts LPS-TLR4 signaling by affecting the expression of *Tlr2* and *Cd14*, activating phosphoinositide 3-kinase, or both"

The reviewer provided good suggestions for studying the relationship between YAP and Tlr, Pik3cb. In our previous study, we already showed that knocking down of Pik3cb attenuated YAP-induced CM proliferation, and that overexpression of Pik3cb in the Yap conditional knockout mice decreased CM apoptosis (Lin et al., 2015). Regarding to YAP and Tlr4/2, we are having following up studies to test the genetic relationship between YAP and Tlr4/2. Nevertheless, these studies are out of the scope of the current manuscript.

Minor/ technical issues:

1. Introduction add more relevant references for Hippo/ Yap involvement in cardiac regeneration.

Response: We put three more sentences into the introduction part on page 4 to describe the roles of Hippo-YAP pathway in cardiac regeneration. "In murine myocardial infarction model, activation of YAP improves cardiac regeneration and reduces myocardial scar size(Xin et al., 2013; Lin et al., 2014). SAV is a scaffold protein that forms complex with Mst1/2 kinase to activate Lats1/2. Inactivation of SAV promotes cardiac regeneration and reverses cardiac systolic function post myocardial infarction". (Heallen et al., 2013; Leach et al., 2017)

2. Figure S2D: images look similar, if you count only the Edu+ CMs. Suggest replacing the images, and/or doing this experiment at 7 days post injury with more EdU injections.

Response: In our previous figure, the 20x images were not clear enough to tell whether the cells were CM or non-CMs. In the revised figure, we replaced the old images with new images, and labeled the EdU+ CMs in the zoom in images.

The reviewer suggests to test the CM proliferation 7 days after IR. However, the expression of modRNA only last for 3 days (Zangi et al., 2013), testing the CM proliferation 7 days after IR will not help to validate whether aYAP modRNA promotes CM proliferation.

3. Figure 3: the legend and figures need some corrections. It's not trivial understanding

what's presented. For example, adding time points (D1/ D2) to A-F would be helpful.

Response: We put more descriptions into the figure legends.

4. Figure 3: analyzing the same heart by FACS and histology is not clear. Variability of sectioning can lead to addition or missing of a highly inflamed region. It might be better to use different hearts for the different experiments.

Response: We appreciate the reviewer's comments.

After IR surgery, the morphology of the IR region was very different from the intact myocardium. When we separated the heart into two parts (base and apex), we were very cautious to make sure the cutting went through the middle of the IR region. The base parts were used for histology study and the apex parts were used for FACS.

Using one heart for two different experiments indeed had a chance to introduce variations. Nevertheless, our histology data and FACS data are consistent, suggesting that this method has enough power to detect the differences in this study. For future studies, to reduce possible variations and increase the detection sensitivity, we will use different hearts for different experiments.

5. The conclusion from the data shown in Figure 3 and Figure S3 is that aYAP reduces innate immune cells infiltration (mainly neutrophils). The way it's written now, it implies that aYAP may affect the neutrophil directly, while the simpler conclusion (shown in Fig. 7I) is that reduced infarct size leads to reduced infiltration, shown by the similar neutrophil density in the infarct of both animal groups. To rule out an initial effect of aYAP on neutrophils, as the modRNA delivery is nonspecific, it would be interesting to look at neutrophil dynamics. Neutrophils are known to be the first cells to infiltrate the injury site.

Response: With the current available data, we can not draw a conclusion how aYAP modRNA reduces neutrophil infiltration. To make this point clear, we added half paragraph in the discussion section on page 16 to address this issue.

The suggestion to do a neutrophil dynamic study is very helpful. We are now developing a nanoparticle-modRNA delivery system to specifically target the neutrophils. With the newly developed tool, we will be able to answer whether aYAP modRNA protects the heart by modulating the neutrophils activity in our following-up studies.

6. Figure 4B, C: the EF values of the sham seems too high. The variation in the injury is very low. What was the sample size for this experiment? Similar to D? Please state it.

Response: . Our EF% measurements were all carried out under awake condition, and the EF% values are in normal range of awake mice (Yang et al., 1999). On the first day after IR, mice with EF% higher than 75% were excluded from this study, which reduced the variation. To clarify this, we put one sentence into the material and methods section on page 5, *Echocardiography*: "Mice with EF% higher than 75% on the first day after IR were excluded from the following studies."

The sample size was stated in the figure legend. Sham, n=4; Veh+IR, n=6; aYAP+IR, n=5.

7. Figure 5C: please specify the GO annotation tool used to do this. GSEA? If so, which module?

Response: GO term signaling pathway enrichment analysis was carried out on the gene ontology resource website (<http://geneontology.org>), using the PANTHER Overrepresentation Test tool. We put this information into the figure legend.

8. Figure 5E: Were the cells treated with the adenovirus as shown in Fig S5? The authors don't refer to it in the legend or text, only at Fig. S5 and 6. Also, the data in Fig 5E and Fig 6H seems contradictory with respect to Tlr4.

Response: We revised the figure 5E legend as "NRVMs were treated with either LacZ or aYAP adenovirus for 48 hours in the absence of serum".

Regarding to Fig. 5E and Fig. 6H, we also noticed the difference when we tested the expression of Tlr4. We reasoned that YAP suppression of Tlr4 may be context dependent. In figure 5E, the NRVMs were only treated with adeno virus, but in figure 6H, the NRVMs were treated with both adenovirus and LPS. Addition of LPS may blunt YAP suppression of Tlr4 expression. We put one sentence on page 13 to explain this phenomenon. "Unlike *Tlr2*, LPS did not induce *Tlr4*. In the baseline condition, YAP suppressed the expression of *Tlr4* (Fig. 5E); however, aYAP did not affect Tlr4 expression in the presence of LPS (Fig. 6H, Suppl. Fig. 5B), suggesting that YAP regulation of *Tlr4* is context dependent."

9. Page 11, first paragraph: when addressing the WB verifying aYAP expression, it should be "Fig. S5B" and not "Fig S5A".

Response: We corrected this error in the text.

Summary:

The novelty of this study is mainly the protective effect of transient Yap activation, and the ability to connect this to TLR signaling and thereby innate immunity. The second claim is still speculative, as it needs a direct evidence. Other technical issues are minor. The transient nature of this delivery system makes it interesting for translational purposes.

We thank the reviewer for the constructive comments. In the current study, we focused on testing the efficacy of aYAP modRNA. In this work, we observed that YAP suppressed CM innate immune response; in our following-up studies, we will use genetic mouse models to test whether YAP suppresses CM innate immune responses by inactivating Tlr4 signaling pathway.

References:

Estarás, C., Hsu, H. T., Huang, L., & Jones, K. A. (2017). YAP repression of the WNT3 gene controls hESC differentiation along the cardiac mesoderm lineage. *Genes Dev*, 31(22), 2250-2263.

Heallen, T., Morikawa, Y., Leach, J., Tao, G., Willerson, J. T., Johnson, R. L. et al. (2013). Hippo signaling impedes adult heart regeneration. *Development*, 140(23), 4683-4690.

- Kim, M., Kim, T., Johnson, R. L., & Lim, D. S. (2015). Transcriptional co-repressor function of the hippo pathway transducers YAP and TAZ. *Cell Rep*, 11(2), 270-282.
- Leach, J. P., Heallen, T., Zhang, M., Rahmani, M., Morikawa, Y., Hill, M. C. et al. (2017). Hippo pathway deficiency reverses systolic heart failure after infarction. *Nature*, 550(7675), 260-264.
- Lin, Z., von Gise, A., Zhou, P., Gu, F., Ma, Q., Jiang, J. et al. (2014). Cardiac-specific YAP activation improves cardiac function and survival in an experimental murine MI model. *Circ Res*, 115(3), 354-363.
- Lin, Z., Zhou, P., von Gise, A., Gu, F., Ma, Q., Chen, J. et al. (2015). Pi3kcb links Hippo-YAP and PI3K-AKT signaling pathways to promote cardiomyocyte proliferation and survival. *Circ Res*, 116(1), 35-45.
- Xin, M., Kim, Y., Sutherland, L. B., Murakami, M., Qi, X., McAnally, J. et al. (2013). Hippo pathway effector Yap promotes cardiac regeneration. *Proc Natl Acad Sci U S A*, 110(34), 13839-13844.
- Yang, X. P., Liu, Y. H., Rhaleb, N. E., Kurihara, N., Kim, H. E., & Carretero, O. A. (1999). Echocardiographic assessment of cardiac function in conscious and anesthetized mice. *Am J Physiol*, 277(5), H1967-74.
- Zangi, L., Lui, K. O., von Gise, A., Ma, Q., Ebina, W., Ptaszek, L. M. et al. (2013). Modified mRNA directs the fate of heart progenitor cells and induces vascular regeneration after myocardial infarction. *Nat Biotechnol*, 31(10), 898-907.
- Zhou, X., Li, W., Wang, S., Zhang, P., Wang, Q., Xiao, J. et al. (2019). YAP Aggravates Inflammatory Bowel Disease by Regulating M1/M2 Macrophage Polarization and Gut Microbial Homeostasis. *Cell Rep*, 27(4), 1176-1189.e5.

August 22, 2019

Re: Life Science Alliance manuscript #LSA-2019-00424-TR

Dr. Zhiqiang Lin
Masonic Medical Research Institutue
2150 Bleecker Street
Utica, NY 13501

Dear Dr. Lin,

Thank you for submitting your revised manuscript entitled "aYAP modRNA reduces cardiac inflammation and hypertrophy in a murine Ischemia-reperfusion model" to Life Science Alliance. The manuscript was assessed by one of the original reviewers again, whose comments are appended to this letter.

As you will see, reviewer #2 expected a better link to altered immune signaling (partially) underlying the protective effect of aYAP (in vitro test overexpressing Tlr2 and/or Cd14), and we outlined in our previous decision letter that such better link is needed for publication here. The reviewer also noted that the EdU images in Fig S2D are not convincing and is disappointed that you did not address this issue in the revision in a better way. The images displayed remain unconvincing in our view.

We usually only allow one round of major experimental revision, but think that you should be able to address the remaining issues in a relatively short time frame by following the suggestions initially made by the reviewer. We would thus like to invite you to re-revise your manuscript, addressing the remaining concerns noted above.

While revising, please also:

- mention the statistical tests used next to the p value in the figure legends
- add a scale bar in Fig S2C
- add Wenqing Cai as an author in our submission system
- clarify authorships please; note that we follow ICMJE authorship guidelines (<http://www.icmje.org/recommendations/browse/roles-and-responsibilities/defining-the-role-of-authors-and-contributors.html>)

Thank you for this interesting contribution to Life Science Alliance. We are looking forward to receiving your revised manuscript.

Sincerely,

B. MANUSCRIPT ORGANIZATION AND FORMATTING:

Reviewer #2 (Comments to the Authors (Required)):

In their rebuttal letter, Chen et al. addressed my comments, and did moderate some of their conclusions to fit a more conservative interpretation of the data. However, some of the suggested comments in my first evaluation suggested additional experiments that were needed to support some of the conclusions stated in this article. The authors choose not to do any of those experiments.

Herein follows the unanswered comments:

Major comments:

I suggested the authors to perform a TLR2 over expression experiment to check the direct interaction between aYAP signaling and the Tlr pathway. While there are some previous work done on that axis (testing Pik3cb), this experiment is still lacking.

Minor comments:

Cardiomyocyte proliferation: As the images describing CM proliferation (S2D) was not conclusive, the authors were suggested to monitor CM proliferation in an extended experiment (7 days) using EdU. The authors did not perform this experiment, as they claim that the aYap modRNA expression is down regulated after 3 days (based on Zangi et al., 2013) and therefore CM proliferation cannot be detected. Reading Zangi et al. reveals that modRNA expression of both reporter gene and VEGF lasts for at least 6 days. Further, CM proliferation have been shown to persist even after the mitogenic signal was turned off. Finally, EdU is a cumulative staining, that will allow the authors to detect proliferation event throughout the 7 days of the experiment, underscoring any difference in proliferation rate, if there is one.

The authors conclude that aYap signaling affect neutrophils directly. One way to establish that is to follow neutrophils dynamics in different time points after aYap treatment, via FACS or other means. The authors did not check that, and rather claim that they will check the direct effect of aYap on neutrophils in a follow up study.

Summary - While the authors did needed text changes they neglected to perform the suggested experiments. I leave it up to the editor to make a decision accept/reject.

Major comments:

I suggested the authors to perform a Tlr2 over expression experiment to check the direct interaction between aYAP signaling and the Tlr pathway. While there are some previous work done on that axis (testing Pik3cb), this experiment is still lacking.

Response: We followed the reviewer's suggestion to test the genetic relationship between YAP and CD14/TLR2 in NRVMs. Although we generated one CD14 adenovirus and one Tlr2 adenovirus, only CD14 adenovirus works (Suppl. Fig. 5F). Tlr2 adenovirus failed to express TLR2 in NRVMs. With Cd14 adenovirus, we were able to prove that CD14 abolished YAP's protection effects in NRVMs (Fig. 6 J-L). These data provide direct evidences that YAP blunts TLR signaling pathway by reducing Cd14 expression.

Minor comments:

Cardiomyocyte proliferation: As the images describing CM proliferation (S2D) was not conclusive, the authors were suggested to monitor CM proliferation in an extended experiment (7 days) using EdU. The authors did not perform this experiment, as they claim that the aYap modRNA expression is down regulated after 3 days (based on Zangi et al., 2013) and therefore CM proliferation cannot be detected. Reading Zangi et al. reveals that modRNA expression of both reporter gene and VEGF lasts for at least 6 days. Further, CM proliferation have been shown to persist even after the mitogenic signal was turned off. Finally, EdU is a cumulative staining, that will allow the authors to detect proliferation event throughout the 7 days of the experiment, underscoring any difference in proliferation rate, if there is one.

Response: For this whole study, we tested whether YAP modRNA had protection effects on the heart after acute IR. When we did the EdU study, the effects of aYAP on EdU incorporation were mild. After IR, aYAP modRNA only doubled the number of EdU+ CMs. It is unlikely that YAP modRNA helps to regrow the myocardium. Therefore, we focused on assessing the roles of aYAP modRNA on cell survival and innate immune response.

The current study mainly talks about cell survival and innate immune. The EdU data set makes the manuscript dispersed, and does not help to draw a conclusion whether aYAP modRNA promotes myocardium regeneration. Therefore, to make this study more focus, and to avoid confusions, we removed the current EdU data set.

The authors conclude that aYap signaling affect neutrophils directly. One way to establish that is to follow neutrophils dynamics in different time points after aYap treatment, via FACS or other means. The authors did not check that, and rather claim that they will check the direct effect of aYap on neutrophils in a follow up study.

Response:

With the current available data, we can not draw a conclusion whether aYAP modRNA directly reduces neutrophil infiltration. To make this point clear, we added half paragraph in the discussion section on page 16 to address this issue in the previous revision:

"In our study, aYAP modRNA significantly reduced CM death, and this was associated with reduced circulating neutrophils and lower myocardial neutrophil infiltration. The reason for aYAP modRNA reducing neutrophils in both the myocardium and the peripheral blood is not clear. One possible explanation is that aYAP modRNA has accumulative effects on reducing DAMPs and cytokines release, and therefore attenuates neutrophil infiltration and bone marrow to peripheral blood mobilization in a time-dependent manner. "

Due to the unspecificity of naked aYAP modRNA, with the current tools, we are unable to dissect the molecular mechanism of how aYAP modRNA affect neutrophil infiltration. We are now developing a nanoparticle-modRNA delivery system to specifically target CMs or neutrophils. With the newly developed tool, we will be able to answer whether aYAP modRNA protects the heart by directly reducing CM death or by directly affecting neutrophils activity in our following-up studies.

Summary - While the authors did needed text changes they neglected to perform the suggested experiments. I leave it up to the editor to make a decision accept/reject.

December 3, 2019

RE: Life Science Alliance Manuscript #LSA-2019-00424-TRR

Dr. Zhiqiang Lin
Masonic Medical Research Institutue
2150 Bleecker Street
Utica, NY 13501

Dear Dr. Lin,

Thank you for submitting your revised manuscript entitled "aYAP modRNA reduces cardiac inflammation and hypertrophy in a murine Ischemia-reperfusion model". As you will see, the reviewer appreciates your final response and we would thus be happy to publish your paper in Life Science Alliance.

Please log into our system one more time, move all files to the new manuscript number and fix the author order within the submission system. Please also fill in the electronic license to publish form.

A. FINAL FILES:

B. MANUSCRIPT ORGANIZATION AND FORMATTING:

Sincerely,

Reviewer #2 (Comments to the Authors (Required)):

In their rebuttal letter, Chen et al. re addressed the comments We set forward, this time performing the needed experiment to clarify the major comment. While not actually performing the experiments

needed to answer the minor comments, I agree with the authors that the main focus of this MS is not on this issues, and the changes they have added to the text are sufficient to allow this paper to be published. Therefore, I see this improved version of the MS as a clearer story, ready for publication.

December 5, 2019

RE: Life Science Alliance Manuscript #LSA-2019-00424-TRRR

Dr. Zhiqiang Lin
Masonic Medical Research Institutue
2150 Bleecker Street
Utica, NY 13501

Dear Dr. Lin,

Thank you for submitting your Research Article entitled "aYAP modRNA reduces cardiac inflammation and hypertrophy in a murine Ischemia-reperfusion model". It is a pleasure to let you know that your manuscript is now accepted for publication in Life Science Alliance. Congratulations on this interesting work.

DISTRIBUTION OF MATERIALS:

Again, congratulations on a very nice paper. I hope you found the review process to be constructive and are pleased with how the manuscript was handled editorially. We look forward to future exciting submissions from your lab.

Sincerely,
